

# SAFENET - Fracture evolution in crystalline rocks (from lab to in-situ scale)

Olaf Kolditz[1,7], Christopher McDermott[2], Jeoung Seok Yoon[3], Jörg Renner[4], Li Zhuang[5], Andrew Fraser-Harris[2], Michael Chandler[2], Samuel Graham[2], Ju Wang[6], and Mostafa Mollaali[1]

[1]Helmholtz Centre for Environmental Research UFZ, Department of Environmental Informatics
[2]School of Geosciences, The University of Edinburgh
[3]DynaFrax UG
[4]Ruhr-Universität Bochum, Institute for Geology, Mineralogy, and Geophysics
[5]Chongqing University, School of Resources and Safety Engineering
[6]Beijing Research Institute of Uranium Geology BRIUG
[7]Technische Universität Dresden

**Correspondence:** Olaf Kolditz (olaf.kolditz@ufz.de)

**Abstract.** The DECOVALEX Task SAFENET is dedicated to advancing the understanding of fracture nucleation and evolution processes in crystalline rocks, with applications in nuclear waste management and geothermal reservoir engineering. Further improvement of fracture mechanics models is required in two distinct areas. Firstly, there is a need to enhance numerical methods for fracture mechanics under varying thermo-hydro-mechanical (THM) conditions. Secondly, there is a requirement

5   to develop applied tools for performance and safety assessment in the context of nuclear waste management, as well as for reservoir optimisation in geothermal applications. Building on the achievements of SAFENET, which concentrated on benchmarking fracture models and experimental laboratory analyses, SAFENET-2 is dedicated to extending and validating models from the laboratory to the field scale.

This paper gives a detailed description of the work plan for SAFENET-2 of the experimental program and the model-

10  ing exercises. The experiments will be carried out at the rock mechanics laboratories of the Universities of Edinburgh and Chongqing. For field data, the STIMTEC experiment at the Reiche Zeche teaching and research mine (Technische Universität Bergakademie Freiberg) is used. The paper gives a detailed description of the individual steps of the task. As a result of SAFENET, the benchmark suite will be made available as interactive exercises via a web portal, thus promoting the concept of open science. The paper will help the teams to organize their work efficiently and also provide an overview and insight to the

15  community.

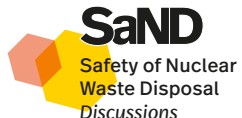

# 1 Introduction

DECOVALEX is a long-term international benchmarking project focusing on the systematic improvement of models for repository research. Multi-physical processes - so-called thermo-hydro-mechanical-chemical (THMC) processes, which can be used to describe the temporal and spatial evolution of repository systems in terms of continuum mechanics - play a special role (Birkholzer et al., 2024). Decovalex is an acronym that stands for "Development of Coupled Models and their Validation against Experiments". The benchmarking philosophy has been continuously refined over numerous project phases. The basic building blocks, which have increasing complexity, are benchmarking exercises, experimental analysis, blind prediction and, increasingly, performance assessments for repository systems. Building trustworthy models is a major research challenge. It is critical to building acceptance in the broader sense (Flynn et al., 1992; Sjöberg, 2004; Sjöberg and Drottz-Sjöberg, 2008). In the context of benchmarking exercises, analytical solutions and/or code comparisons are employed for the purpose of academic, synthetic test examples. These are used to ascertain the accuracy of numerical models and/or to test the correct implementation of numerical methods. In the case of Decovalex tasks, experiments from geotechnical and underground laboratories are selected on a systematic basis with a view to validating the numerical models against measured data. The question of the transferability of the models from the laboratory to the in-situ scale plays a central role in this process (experimental analysis). In this context, the term 'validation' refers to the capacity of models to predict measurement outcomes that were not incorporated into the calibration process (blind prediction). In recent Decovalex projects, tasks for the characterisation of parts or complete repository systems in a geological context have also been defined and processed (performance assessment). The complete analysis of a repository system requires the handling of a significant computational burden. Consequently, the development of efficient computing methods, particularly parallel computing, is becoming a crucial aspect of Decovalex. Additionally, alternative approaches are being explored to identify suitable replacement models for the intricate coupled THM models utilising machine learning techniques (Bang et al., 2020; Hu et al., 2023; Hu and Pfingsten, 2023; Buchwald et al., 2024; Hu et al., 2024).

Notwithstanding the long history of Decovalex (Rutqvist et al., 2005; Chan et al., 2005), the SAFENET Task introduces fracture mechanics into the project for the first time in a comprehensive benchmarking exercise. SAFENET has a broader perspective, the abbreviation stands for "Safety Assessment of Fluid Flow, Shear, Thermal and Reaction Processes within Crystalline Rock Fracture Networks". To achieve SAFENET's first scientific goal of better understanding fracture initiation and evolution in crystalline rocks under hydro-mechanical and thermo-mechanical loading, a systematic experimental and modeling program was organized and completed under Task G of Decovalex 2023 (experimental analyses). Three experimental programs were carried out in Freiberg (Frühwirt et al., 2021), Seoul (Sun et al., 2021, 2023) and Edinburgh (McDermott et al., 2018; Fraser-Harris et al., 2020, 2023) to study mechanical and thermo-mechanical shear and stress-dependent permeability changes in fractured crystalline rocks. For numerical analyses, the modeling teams offer a wide range of continuum mechanics and discontinuous methods for the numerical modeling of fracture mechanics processes. Details of the numerical methods are described in Mollaali et al. (2023) and Bilke et al. (2019). The benchmark exercises include plane and rough fracture examples as well as simple fracture networks.



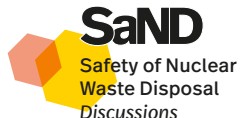

SAFENET will also elaborate the potential of Artificial Intelligence (AI) concepts for benchmarking purposes regarding the use of surrogate models for computational speed-up, quantification of uncertainties (Kurgyis et al., 2024), etc. AI methods are also gaining increasing attention in the field of nuclear waste management (BASE, 2023; Breitkreutz et al., 2023) and particulary in the context of European initiatives such as EURAD (Claret et al., 2022; Prasianakis et al., 2020; Kolditz et al., 2023; Jacques et al., 2023; Kühn et al., 2012). However, the potential needs to be carefully assessed and exploited through
concerted action.

## 2   SAFENET-2

### 2.1   Concept

Figure 1 illustrates the concept of SAFENET-2. In light of the findings presented in Decovalex 2023 regarding the examination of HM and TM processes, the subsequent phase of the investigation of fully coupled THM processes will focus on two distinct
approaches, both commencing from HM (GREAT cell) and TM-H (Thermoslip-flow cell) processes. In a series of previous works (Sun et al., 2021, 2023, 2024b, a), the experimental basis for TM-H was developed. These approaches will encompass the analysis of temperature and hydraulic effects, respectively. The experimental programme has been designed with these objectives in mind. The second area of focus is the transfer of knowledge from the laboratory to the field scale. The experimental basis at the field scale is provided by the STIMTEC experiment at the research mine "Reiche Zeche", where stimulation tests
with periodic pumping tests and high-resolution seismic monitoring have been conducted (Boese et al., 2021, 2022, 2023). In conjunction with the laboratory experimental data, the STIMTEC experiment will serve as a foundation for upscaling fracture models from the laboratory to the field scale with respect to hydro-mechanically induced fracture processes. The SAFENET-2 project incorporates a methodological phase. In this context, further development will be undertaken of numerical approaches to fracture mechanics, including those based on THM (e.g. phase field methods, discrete element methods, etc.). The potential
of Artificial Intelligence (AI), including machine learning methods for the construction of surrogates for complex THM fracture mechanics models will be investigated. Such surrogates may be trained from full complexity THM models. Moreover, novel benchmarking techniques will be introduced that facilitate interactivity in collaborative endeavours through the utilisation of web-based Jupyter notebooks for online benchmarking.

Participating groups of SAFENET-2 are: Helmholtz Centre for Environmental Research (UFZ), Ruhr University Bochum
(RUB), Federal Institute of Geosciences and Natural Ressources (BGR), Technische Universität Bergakademie Freiberg (TUBAF), Chinese Academy of Sciences (CAS), Lawrence Berkeley National Laboratory (LBNL), Sandia National Laboratory (SNL), Edinburgh University, DynaFrax, Chongqing University (CQU), Korean Institute for Geosciences and Mineral Ressources (KIGAM), and Taipower (TPC).



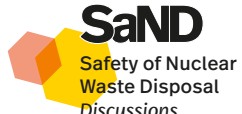

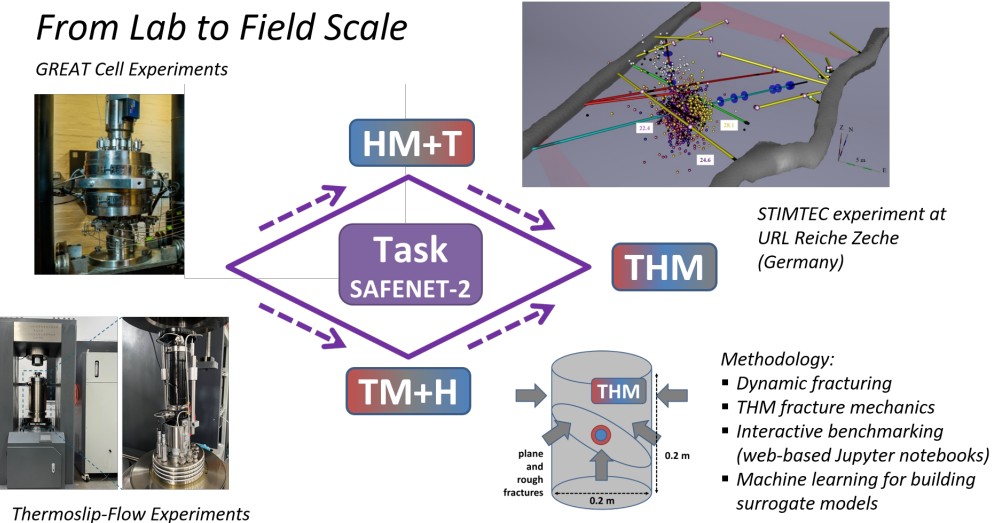

**Figure 1.** Concept of the SAFENET-2 task of Decovalex 2027 following two routes towards THM processes and upscaling models from lab to field scale

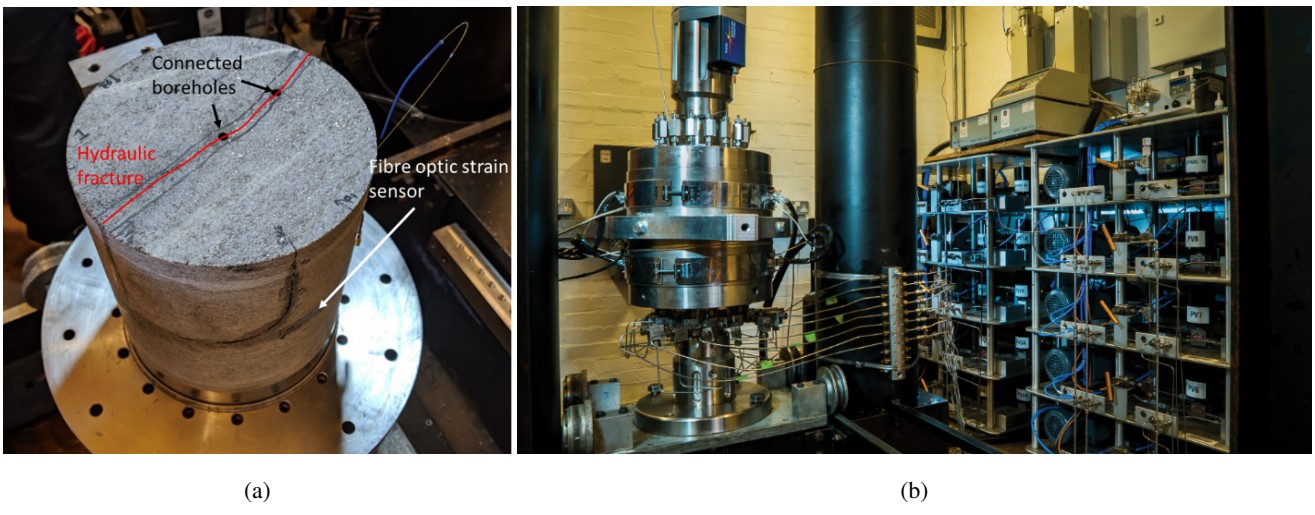

(a)             (b)

**Figure 2.** GREAT cell facility: (a) Freiberg gneis sample after the fracture stage, the orientation of the foliation is highlighted along with pre-existing sealed fracture that is interpreted as the youngest (and potentially weakest) discontinuity in the sample; (b) experimental apparatus

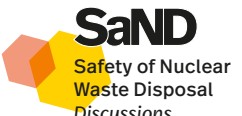

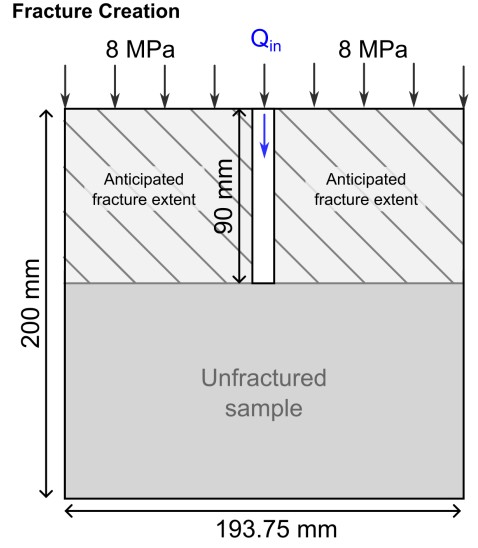

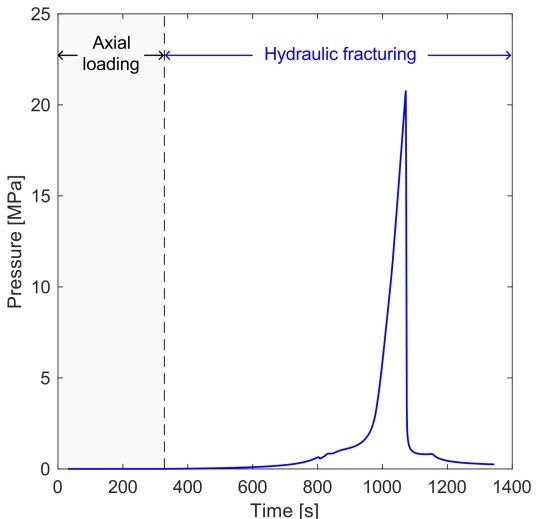

(a) Schematic cross-section of the fractured cylindrical sample

(b) Fluid injection pressure recording during hydraulic fracturing

**Figure 3.** GREAT cell fracture initiation experiment: SAFENET-2-FE

## 2.2 Experimental basis

The experimental basis of Safenet-2 consists of laboratory experiments at University of Edinburgh in UK (GREAT cell facility, section 2.2.1) and Chongqing University in China (section 2.2.2). Experimental data from field experiments are formed from the teaching and research mine of Technische Universität Bergakademie Freiberg in Germany (section 2.2.3).

### 2.2.1 GREAT cell large lab scale data

The GREAT cell facility at University of Edinburgh (Fig. 2) provides the unique capability to create hydraulic fractures in
rock samples under a controlled true-triaxial stress field ($\sigma_1 > \sigma_2 > \sigma_3$), and to change that stress field during the experiment enabling the investigation of the impact of normal and shear stress on fracture permeability. The sample size is 200 mm diameter x 200 mm height, and strain is measured along the middle circumference of the cylinder. The strain is recorded using a fiber optic cable attached to the surface of the sample, allowing a high spatial (every 2mm) and temporal (100 Hz) resolution of the strain to be recorded and the deformation during the process of fracturing to be recorded.

Two types of main experiments are available for the SAFENET-2 HM Task 2: (i) fracture creation and dynamic propagation experiments (three different loading conditions) and (ii) fracture circulation experiments.

**(i) Fracture initiation experiments:** Three 20cm diameter cylindrical rock samples were manufactured, i.e. Freiberg Gneis (one sample), and G603 Granite (two samples). These samples were each used to conduct fluid injection experiments under three distinct stress-states. For the Freiberg Gneis an unconfined press has been used to provide the axial load and an unconfined



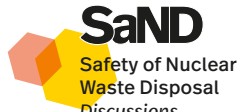

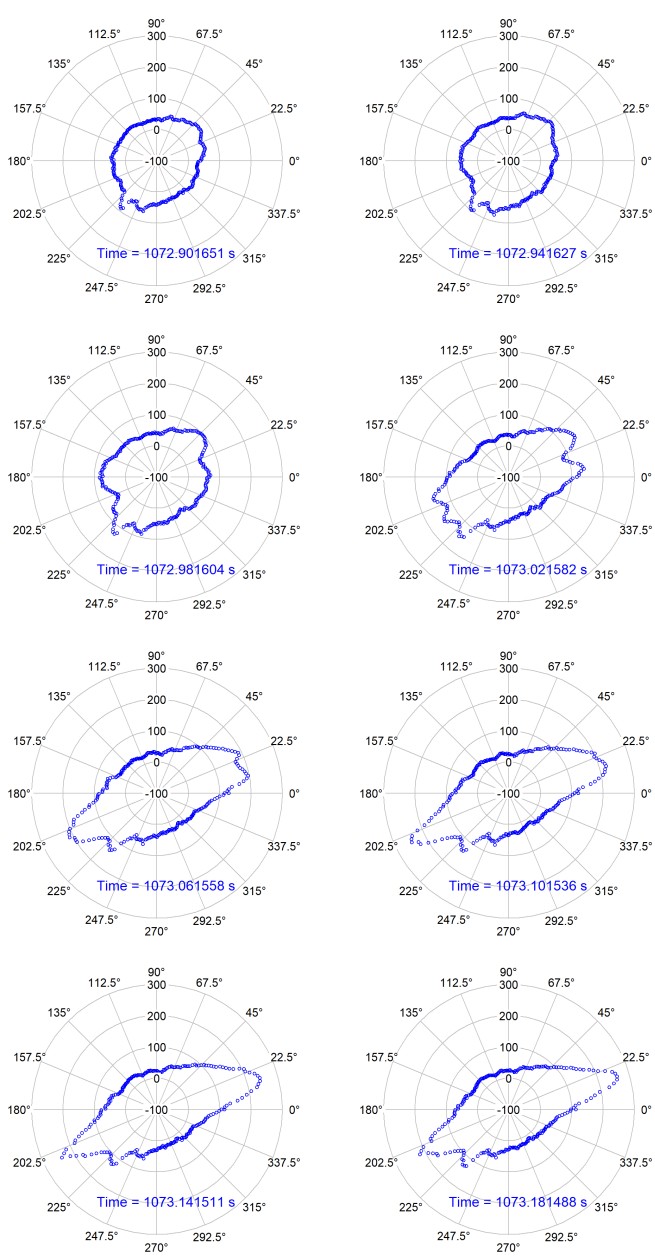

**Figure 4.** Circumferential strain measurements beginning from $t = 1072.901651$ and $\Delta t \approx 0.04$ second intervals during hydraulic fracturing of the Freiberg Gneis sample. Given times in the individual subfigures correspond to the official time stamps of the experiment.

hydraulic fracture has been created using fluid pressure in a borehole. The LUNA fibre-optic strain gauges[1] and logging equipment were used to measure the dynamic circumferential strain during the fracturing process. A schematic of the sample

---

[1]https://lunainc.com/capability/strain

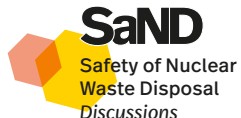

and borehole with respect to the applied stresses is illustrated in Fig. 3a. The evolution of borehole fluid pressure during the fracturing of the Freiberg Gneis sample under uniaxial compressive strength is shown in Fig. 3b. The early part of this plot corresponds to the application of an 8 MPa axial load during which time no increase in borehole fluid pressure was applied.

Fracturing was achieved by flowing water at a constant rate of $Q_i = 1$ ml min$^{-1}$ into the borehole. Inspection of the sample post-fracturing showed that the generated fracture cut the full diameter of the sample, however it did not propagate the full height of the sample. The anticipated extent of the generated fracture shown in Fig. 3 is inferred from these observations. The deformation around the edge of the sample was recorded in terms of circumferential strain at 0.04 s intervals (25 Hz) during the fracturing process (Fig. 4). In addition, two dynamic fracture experiments and rotational deformation tests are available for the

teams to model in granite under radially symmetric loading conditions of 8 MPa and 12 MPa with corresponding axial loads of 24 MPa and 36 MPa. The dynamic deformation is recorded using the LUNA fibre optic cable, the strain record is complete for the 12 MPa experiment, part of the record is missing for the 8 MPa experiment.

**(ii) Fracture circulation experiments:** This experiment is designed to examine fluid flow in a fracture under a combination of different stress orientations and stress anisotropy ratios. Fluid flow through the fracture was facilitated by a second borehole

drilled into the fracture developed within the Freiberg Gneis during the previous unconfined initiation experiments. Fig. 5a shows the schematic diagram of the fluid flow arrangements during the fracture circulation experiments. The extent of the hydraulic fracture generated during the uniaxial experiment is depicted in dark green, with the extend of the fracture available to fluid circulation in light blue. During the experiments fluid was injected into the original borehole in the centre of the sample and allowed to leave the sample via a second borehole positioned radially 50 mm away from the injection borehole. The fluid

injection rate was $Q_i = 5$ ml min$^{-1}$, with fluid viscosity $\mu = 1.03 \times 10^{-3}$ Pa s (close to the viscosity of pure water). Fluid was supplied via a pair of syringe pumps operating in constant flow-rate mode.

After initial fracture creation experiments, two different stress rotation experiments were conducted. Rotation of the stress field around the sample with respect to the fracture allows us to investigate the relative impact of shear and normal stresses acting on the fracture surface on the fluid flow characteristics of the sample. The loading conditions for the GREAT cell

pressure exerting elements (PEEs) employ opposing banks of PEE triplets to apply $\sigma_2$ and $\sigma_3$. These are separated by single pairs of PEEs with a bridging stress, $\sigma_{\text{Bridge}}$, defined as the average of the intermediate and minimum principal stresses (Fig. 5b). Circumferential strains were measured continuously through rotation experiments to assess the influence of shear stress and stress orientation.

In the first stress rotation experiment a $\sigma_2$ and $\sigma_3$ were fixed at 12 MPa and 6 MPa respectively, and rotated in 8 stages by

22.5° steps (stages 0 and 8 are identical in terms of applied stresses), with a bridging stress of 9 MPa (Fraser-Harris et al., 2020). The corresponding circumferential strains are shown in Fig. 6. The large blue arrows indicate the orientation of $\sigma_2$ (maximum horizontal stress), and the small blue arrows indicate the orientation of $\sigma_3$ (minimum horizontal stress) with respect to the fracture. $\sigma_1$ is the axial stress. The stresses in each PEE pair during rations are given in Table 1, where $\sigma_2 = \sigma_{2\text{Max}}$.

In the second rotation experiment the protocol was extended to include a series of sub-stages during each stress rotation

whereby $\sigma_2$ is progressively increased from an initial axisymmetric stress state ($\sigma_2 = \sigma_3$) to $\sigma_2 = \sigma_1$ to investigate the impact of stress anisotropy. Before each incremental rotation of the stress field, a final sub-stage returns the sample to the initial

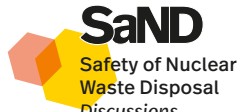

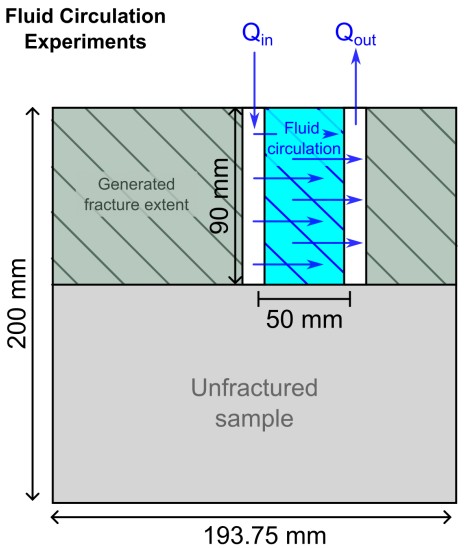

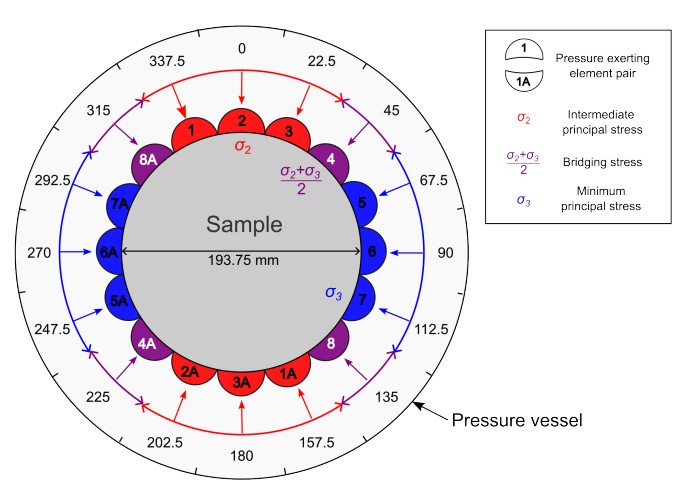

(a) Schematic of the fractured sample for the fluid circulation experiment

(b) Schematic of the rotating stress field application via PEE (pressure exerting elements) triplets

**Figure 5.** GREAT cell circulation experiment: SAFENET-2-CE

**Table 1.** The radial stress conditions employed in the fracture circulation experiments. The axial stress, $\sigma_1$, is maintained at 12 MPa throughout the experiment. Sub-stages increment $\sigma_2$ from $\sigma_{2\mathrm{Min}} = 6$ to $\sigma_{2\mathrm{Max}} = 12$ MPa in 1 MPa increments. The bridging stress $\sigma_{\mathrm{Bridge}} = (\sigma_2 + \sigma_3)/2$, is incremented in steps of 0.5 MPa from 6 to 9 MPa. The column headed *"stress"* refers to the assignment of principal and bridging stresses prior to any rotations.

| PEE Pair | Stress$_{\theta=0}$ | Rotational Step Number (Angle, $\theta$) | | | | | |
| --- | --- | --- | --- | --- | --- | --- | --- |
| | | 0 (0°) | 1 (22.5°) | 2 (45°) | ⋯ | 7 (157.5°) | 8 (180°) |
| **Axial Load** | $\sigma_1$ | 12 | 12 | 12 | | 12 | 12 |
| **1 & 1A** | $\sigma_3$ | 6 | 6→9 | 6→12 | ⋯ | 6 | 6 |
| **2 & 2A** | $\sigma_3$ | 6 | 6 | 6→9 | ⋯ | 6 | 6 |
| **3 & 3A** | $\sigma_3$ | 6 | 6 | 6 | | 6→9 | 6 |
| **4 & 4A** | $\sigma_{\mathrm{Bridge}}$ | 6→9 | 6 | 6 | **Steps** | 6→12 | 6→9 |
| **5 & 5A** | $\sigma_2$ | 6→12 | 6→9 | 6 | **3 to 6** | 6→12 | 6→12 |
| **6 & 6A** | $\sigma_2$ | 6→12 | 6→12 | 6→9 | | 6→12 | 6→12 |
| **7 & 7A** | $\sigma_2$ | 6→12 | 6→12 | 6→12 | ⋯ | 6→9 | 6→12 |
| **8 & 8A** | $\sigma_{\mathrm{Bridge}}$ | 6→9 | 6→12 | 6→12 | ⋯ | 6 | 6→9 |



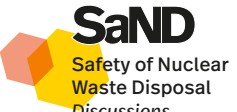

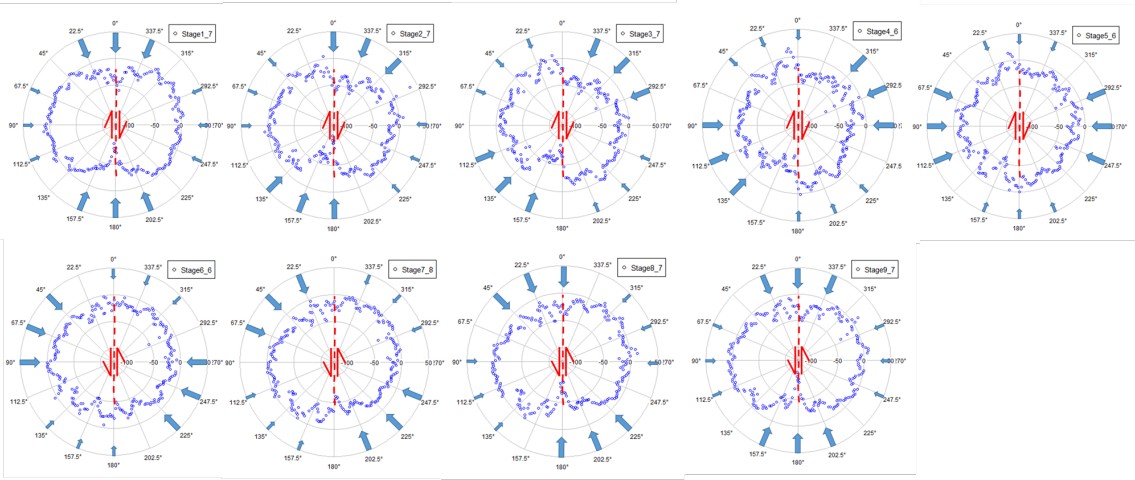

**Figure 6.** Circumferential strains during the first stress rotation experiment. Maximum stress, $\sigma_1$ = 12 MPa and mimimum stress, $\sigma_3$ = 6 MPa. Intermediate stress $\sigma_2$ = 12 MPa is rotated around the sample in 8 stages. In this case the results have not been normalised to stage 3_1 Fig. 7

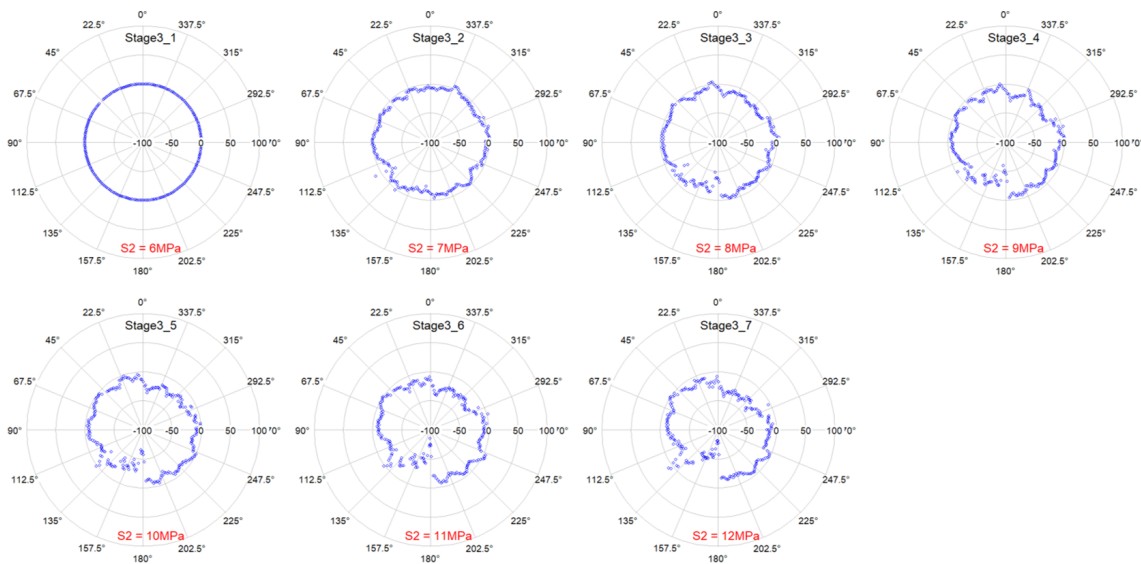

**Figure 7.** Circumferential strains during the second stress rotation experiment, with increasing $\sigma_2$ stresses around 135°-315°axis, $\sigma_3$= 6 MPa $\sigma_1$ = 12 MPa in all cases, all normalised to stage 3_1

axisymmetric stress state at the start of the rotation. Fig. 7 shows the influence of increasing shear stress on fracture and surface strains. Increasing stresses $\sigma_2$ centered around the 135°-315°axis were applied in stages from 6 to 12 MPa. As can be seen, the



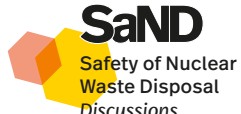

compressive strains decrease in response to progressive increases in $\sigma_2$, whereas dilational strains increase in the orthogonal
direction (45°-225° axis).

Fracture permeabilities have been estimated from both stress rotation experiments. No direct measure of the hydraulic
aperture $b$ [L], of the fracture was possible during either stress rotation experiment. In both experiments, the inferred planar
nature of the fracture between the injection and and fluid return boreholes justified the use of the cubic law for fracture
permeability (Eq. 1a) from which the hydraulic aperture could be estimated;

$$Q_i = \frac{wb^3}{12\mu}\frac{\Delta P}{L} \tag{1a}$$

and consequently the intrinsic permeability, $k$ [L$^2$], of the fracture could be obtained;

$$k = \frac{b^2}{12} \tag{1b}$$

Additionally, $Q_i$ [L$^3$ T$^{-1}$], is the fluid injection/return rate, $L$ [L], is the separation of the injection and return boreholes, $\mu$
[M L$^{-1}$ T$^{-1}$] is the fluid viscosity, and $w$ [L], is the vertical height of the fracture. $\Delta P$ [M L$^{-1}$ T$^{-2}$] is the applied pressure
differential. Throughout all rotations the injection pumps injected at the required pressure to maintain the prescribed flow rate,
against a fixed downstream pressure of 4.89 MPa as set via a backpressure regulator fitted to the return fluid line.

Fig. 8a shows the estimated permeabilities from the experiments where $\sigma_2$ was incrementally increased during each rotation,
as a function of mean modelled normal stress Permeability is seen to decrease with increasing normal stresses in the plane of
the fracture. Likewise, Fig. 8b shows the modelled maximum shear stress in the plane of the fracture. Again permeability
appears to decrease with increasing maximum resolved shear stress, however the sensitivity in change in permeability appears
to be more pronounced than for the resolved normal stress.

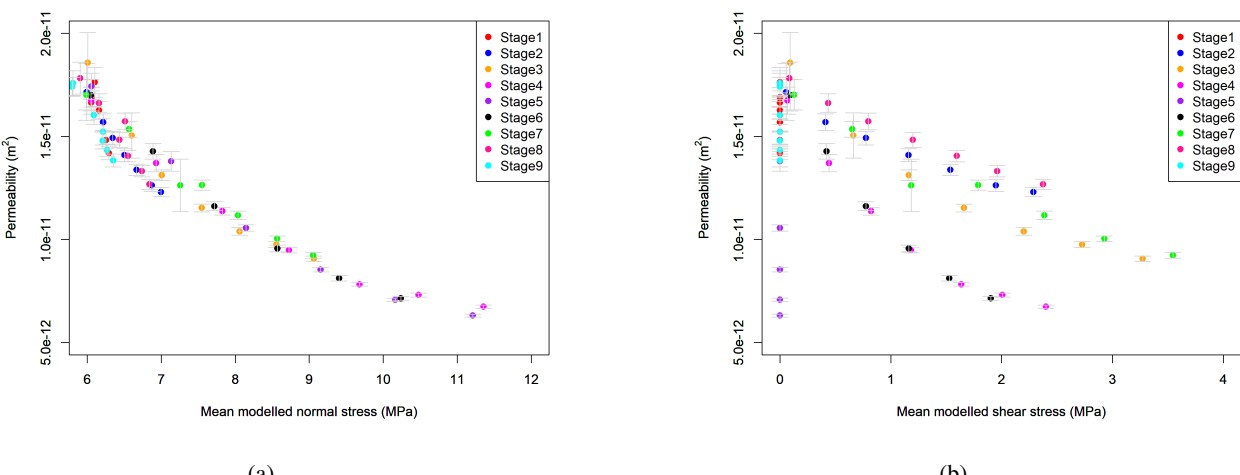

(a)                                              (b)

**Figure 8.** Variation of fracture permeability as a function of normal and shear stresses



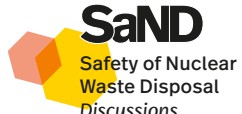

### 2.2.2 Thermoslip-flow lab scale data

The concept of thermoslip-flow test is illustrated in Fig. 9. Using the customized fault shear flow testing machine at Chongqing University, an inclined natural rock fracture with roughness cutting through a cylindrical rock specimen (50 mm diameter $\times$ 100 m length) is initially loaded close to criticality. The specimen is then heated by heating the confining oil. During the heating period, $\sigma_3$ will be constant while thermal expansion is restricted in the axial direction. As a result, thermal stress ($\sigma_T$) will be generated and added to $\sigma_1$, as shown in Fig. 9a. Thermoslip occurs when the $\sigma_1$ is high enough to reach the Mohr-Coulomb failure criterion.

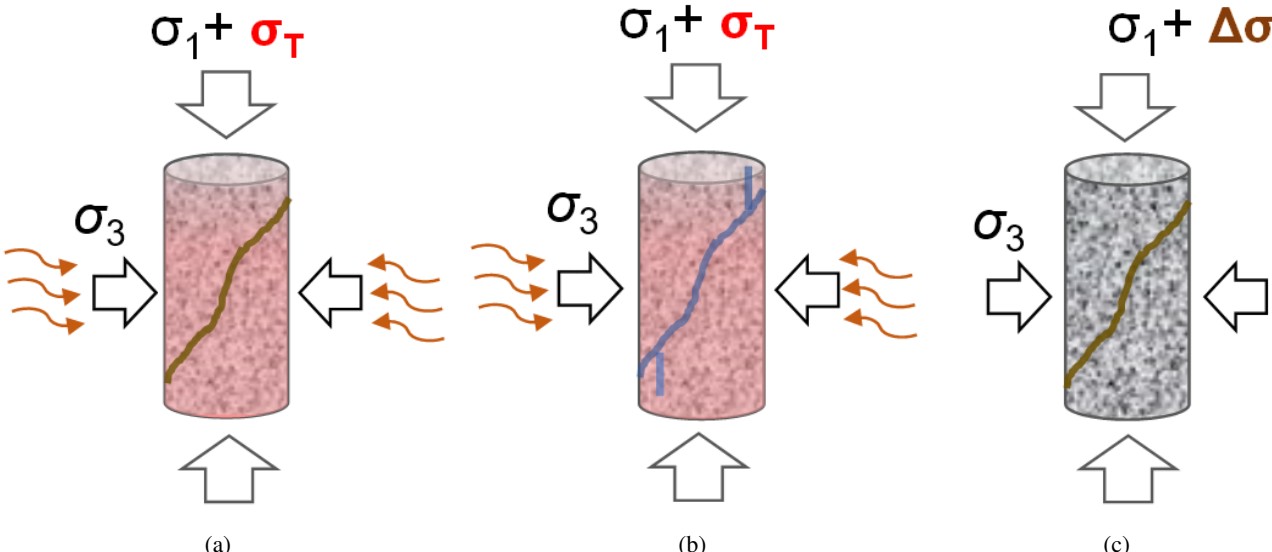

**Figure 9.** Schematic diagram of the experiments: (a) thermoshearing test (or thermoslip test), (b) thermoslip-flow test for the modeling benchmarking, and (c) mechanical shear test.

It is commonly known that shear dilation of rough fractures will cause permeability increase. Therefore, evaluation of hydraulic performance before and after the thermally induced slip is planned. This was not considered in the last phase of SAFENET, where all the thermoshearing test was conducted under a dry condition (Sun et al., 2021, 2023, 2024a). Fluid is injected through the fluid inlet at the bottom and flows through the fracture to the fluid outlet, as shown in Fig. 9b. The pressures at the inlet and outlet are monitored. The fluid flow characteristics before and after thermally induced fracture slip are analysed. In this way the coupled TM+H behaviour of a rock fracture is studied. The data obtained from the thermoslip-flow test in Fig. 9b will be used for numerical modelling benchmarking.

In addition, the thermoshearing test (TM) without fluid interaction and mechanical shear test (M) without thermal or fluid interaction are planned for comparison of shear behaviour, as shown in Fig. 9a and 9c. To ensure comparability, we plan to



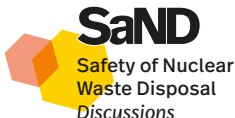

use a reproducible rough fracture with nearly the same topography in the M, TM and TM+H experiments. The fracture will be produced using a digital stone engraving machine with a maximum location accuracy of 10 $\mu$m.

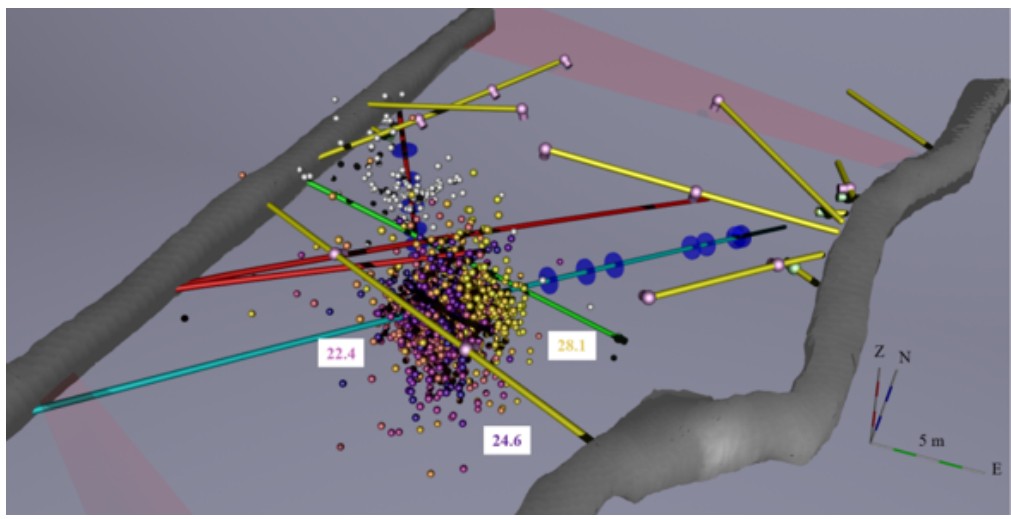

**Figure 10.** STIMTEC field experiment in the underground teaching and research mine "Reiche Zeche" at Technical University Freiberg

### 2.2.3 STIMTEC in-situ experiments

The experimental basis at field scale is provided by the STIMTEC experiment in the teaching and research mine Reiche Zeche where stimulation tests with periodic pumping tests and high-resolution seismic monitoring have been conducted (Boese et al., 2021, 2022; Blanke et al., 2023; Boese et al., 2023). Statistical properties for the characterization of the stress field heterogeneity have been analyzed by Jimenez-Martinez and Renner (2023). Investigations of the main hydro-mechanical phenomena and characteristics of the in-situ experiment have been carried out by (Schmidt et al., 2021, 2023) also within the GeomInt project (Kolditz et al., 2021). Together with laboratory experimental data, the STIMTEC experiment will provide a basis for upscaling fracture models from laboratory to field scale with respect to hydro-mechanically induced fracture processes.

### 2.3 Modelling approach - Steps of the DECOVALEX Task

All DECOVALEX tasks are organized in steps. The SAFENET task is divided into two groups (i) conceptual work (Steps 1 and 5) and (ii) experimental analyses of lab and field experiments (Steps 2, 3 and 4):

- Step 1: Benchmarking (section 2.3.1)

- Step 2: GREAT cell experiments (section 2.3.2)

- Step 3: Thermoslip experiments (section 2.3.3)

- Step 4: STIMTEC experiments (section 2.3.3)



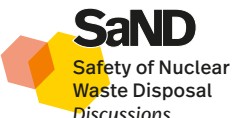

– Step 5: Synthesis (section 2.3.4)

### 2.3.1  Step 1: Benchmark simulations

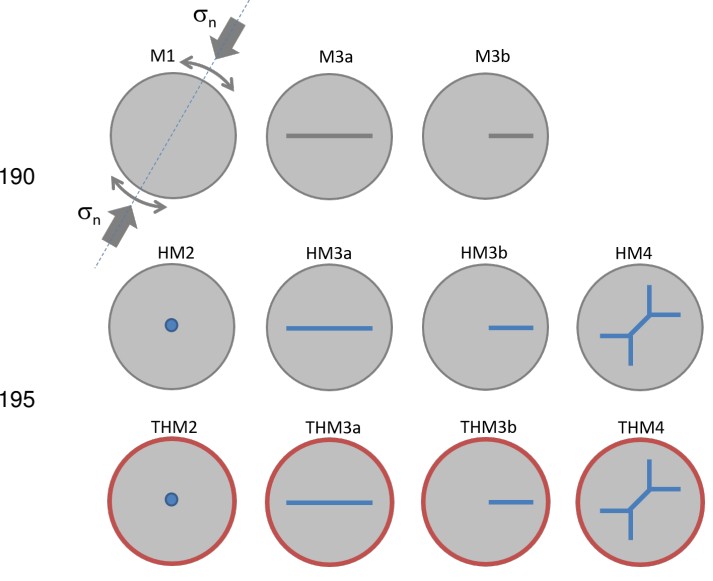

The main objective of Step 0 is to provide a suite of benchmarks for SAFENET-2. These benchmarks will cover the basic features of the GREAT cell, Chongqing and STIMTEC experiments in a simplified way so that the modeling teams can test their methods and code to see if they are in principle able to simulate the laboratory and in-situ experiments. The benchmark suite will therefore provide a common basis for the modeling teams and allow typical benchmark exercises such as grid convergence tests to prove correct discretizations for simulating the fracture processes with sufficient accuracy. The benchmark suite will be made available as an open science contribution via interactive Jupyter notebooks to encourage more teams not currently involved in Decovalex to participate in these benchmarking exercises and to create an easily findable, accessible, interoperable, and re-

**Figure 11.** GREAT cell THM benchmarking suite

producible reference.

**Step 1.1**: Fig. 11 shows the benchmark suite for THM fracture processes that is featured by the GREAT cell experiments - but also serving for benchmarking TM and TM+H models for the thermoslip experiments. The basic idea is to mimic fracture

processes in a rotating stress field. The HM version has already been completed as part of SAFENET-1 (Mollaali et al., 2023). The THM includes thermal processes by externally heating the rock samples to mimic different thermal boundary conditions (e.g. depth-dependent geothermal temperatures).

**Step 1.2**: Thermoslip-flow benchmarks)

The modeling exercises are divided into the following sub-steps according to the main processes: M (Fig. 12a), TM (Fig.

12b), HM (Fig. 12c) and THM (Fig. 12d) experiments. Since the main features of the experiments can also be simulated in plane-strain models, we start the modeling exercises in 2D and then continue in 3D in Step 3 (Fig. 16b). Additionally, models for both plane and rough fractures will examined.

**Step 1.3**: Fig. 13 shows the concept of the STIMTEC benchmark - a fracture embedded in a gneis rock block. The fracture is intended to represent one of the larger discontinuities observed in the research mine (i.e. realistic fracture orientation). Realistic

in-situ stresses are imposed by appropriate stress boundary conditions. The basic idea of the is benchmark test is to qualitatively reproduce the hydraulic pressure responses to the various stimulation phases during the STIMTEC experiment, i.e. p-test, frac test, re-fracs, step-rate test, shut-in and periodic pumping tests (see sec. 2.3.3).



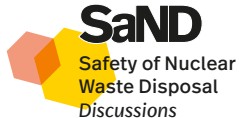

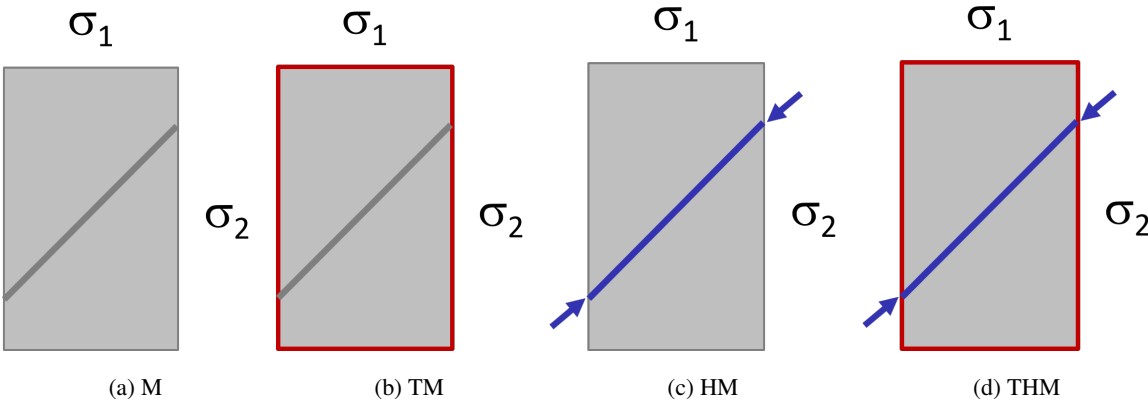

| (a) M | (b) TM | (c) HM | (d) THM |

**Figure 12.** Benchmark exercises for the thermoslip-flow cell experiments: investigation of the individual mechanical (a) and mechanical coupled processes with increasing complexity (b-d).

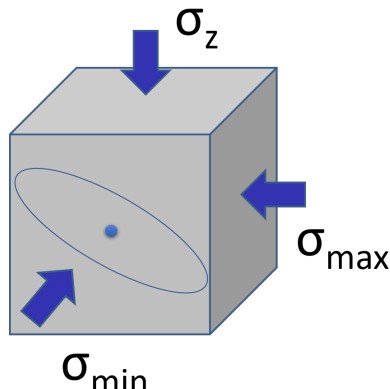

**Figure 13.** STIMTEC benchmark concept

### 2.3.2   Step 2: GREAT cell test case simulation

Three experiments conducted with Gneis and Granite samples are available for simulation. A "large" natural heterogeneous
foliated Freiberg Gneis sample is hydraulically fractured under axial load confined conditions, and unconfined radial conditions
– fracture is free to form according to the influence of foliations:

- Process Dynamic Fracture Formation, unconfined
- Material: Freiberg Gneis
- Conditions : $\sigma_1 = 8$ MPa, $\sigma_2 = \sigma_3 = 0$, $\varnothing = 200$ mm



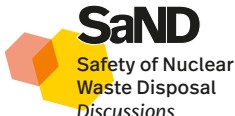

– Experimental Data : Time dependent fluid pressure, surface deformation @100Hz, dynamic fracture growth recorded

Two "large" Granite samples are hydraulically fractured under axial load confined conditions, and confined radial conditions. Two boreholes are drilled into the fracture to facilitate fluid flow measurements, and estimates of permeability under different stress and fluid flow conditions are available:

  – Process Dynamic Fracture Formation, confined x2

– Material: G603 Granite

  – Conditions : $\sigma_1$ = 24 MPa, $\sigma_2 = \sigma_3$ = 8 MPa, $\varnothing$ = 200mm

  – Conditions : $\sigma_1$ = 32 MPa, $\sigma_2 = \sigma_3$ = 12 MPa, $\varnothing$ = 200mm

  – Experimental Data: Time dependent fluid pressure, surface deformation @100Hz, dynamic fracture growth recorded.

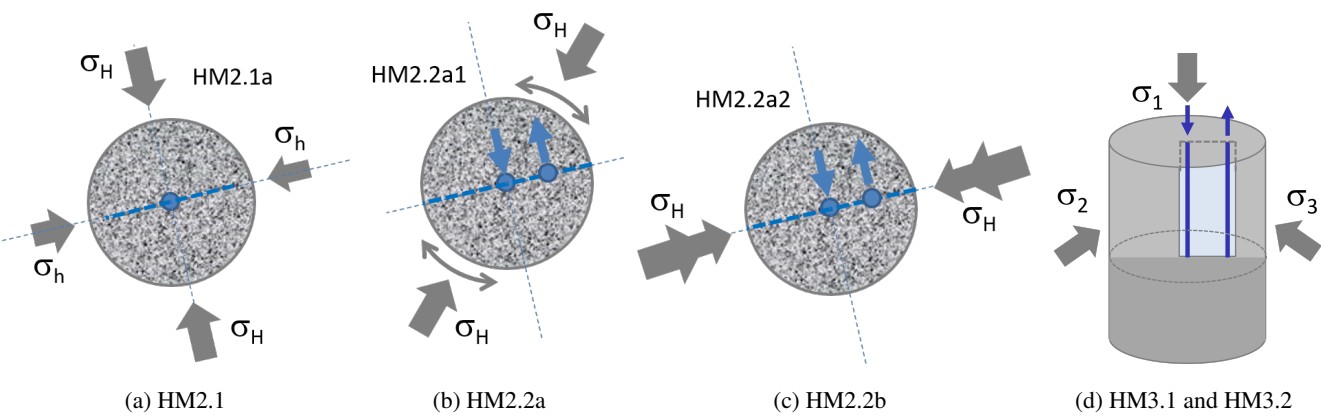

(a) HM2.1          (b) HM2.2a          (c) HM2.2b          (d) HM3.1 and HM3.2

**Figure 14.** Modelling exercises for the GREAT cell experiments

The modeling exercises are divided into the following substeps according to the two main experiments, hydraulic fracturing
(substep 2.1) and flow circulation (substep 2.2). In substep 2.1, the hydraulic fracturing process is simulated according to the experimental conditions (Fig. 14a). In substep 2.2 the flow circulation experiment will be modeled in two versions according to the rotating stress field (substep 2.2a, Fig. 14b) and the stepwise increase of stress in the direction of the fracture orientation (substep 2.2b, Fig. 14c). Since the main features of the experiments can also be simulated in plane-strain models, we will start the modeling exercises in 2D and continue them in 3D (Fig. 14d).

### 2.3.3  Step 3: Thermoslip-flow test case simulation

Granite cores from the Beishan underground research laboratory are planned to be used in the experiments. For details about the physical and mechanical properties we refer to Chen et al. (2023); Yi et al. (2024). The experiment team in Chongqing



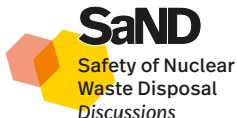

University will select a granite core containing a typical rough fracture or natural fracture, as shown in Fig. 15. The surface topography of the fracture will be obtained using a high resolution optical 3D scanner, and the initial contact conditions between

the top and bottom can be investigated using a CT scan, as shown in Fig. 15a. In situ stress and heating conditions will be decided based on a pre-modeling, which will give details about distributions of temperature and thermal stress inside the granite specimen (Sun et al., 2024b). The surface profile data of the fracture, basic physical and mechanical properties of the granite, and heating boundaries will be delivered during the 2$^{nd}$ DECOVALEX workshop in October 2024. With the specimen properties and given heating conditions, benchmarking teams shall conduct numerical modeling to estimate temperature, thermal stress

and shear behavior of the rough fractures. This process is defined as a blind prediction. The keywords of the test results include but not limited to) asperity damage, (2) interlock, (3) slip pattern and (4) permeability change. Suggested items on the benchmarking list are:

- Temperature and thermal stress distribution
- Slip displacement and slip pattern
- Shear dilation and permeability change
- Asperity damage distribution
- Influence of interlock

In particular, we are curious about effect of interlock on the slip behavior such as slip pattern and shear dilation, as illustrated in Fig. 15b. The Step 2 test case simulation will commence in late 2025. More information will be available and distributed to

the research team during the 2$^{nd}$ DECOVALEX workshop in October 2024, and the 3$^{rd}$ DECOVALEX workshop in May 2025.

Since the main features of the experiments can also be simulated in plane-strain models, we start with the benchmark exercises in 2D and then continue in 3D (Fig. 16). Additionally, models for both plane and rough fractures will examined.

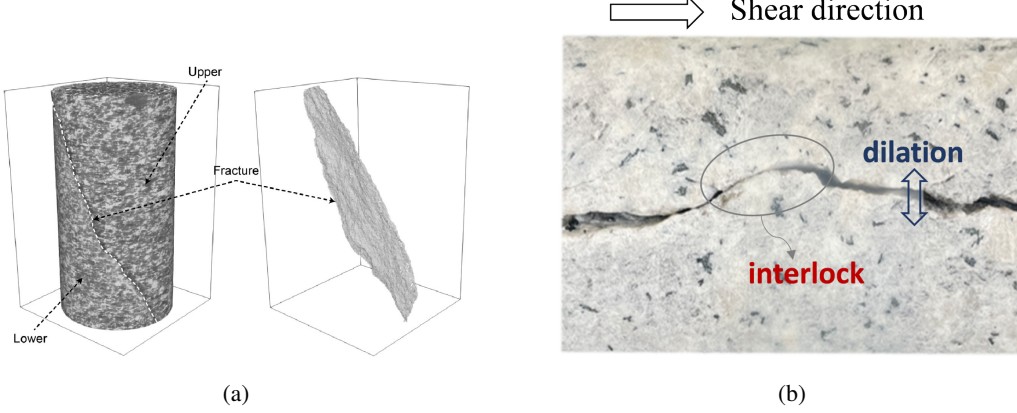

(a)                 (b)

**Figure 15.** Granite specimen used for experiments: (a) the inclined through-going rough fracture in a granite cylinder, (b) an example showing the upper fracture climbed over the lower one in an originally interlocked granite specimen. Shear dilation occurred as a result of shearing.

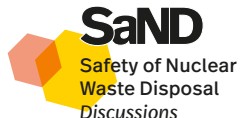

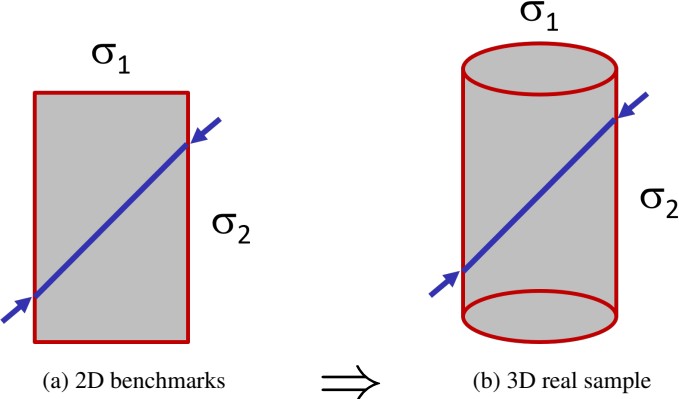

**Figure 16.** Extending thermoslip-flow cell models from 2D benchmarks to 3D real samples

**Step 4: STIMTEC in-situ experiments**

Hydraulic injection and stimulation results will be analyzed to characterize in situ fracture permeabilities. Acoustic emission
data will be used to constrain fracture mechanics models. Fig. 17 shows the pressure response to the subsequent hydraulic test
stages:

(i) p-test (ii) Hydraulic fracturing (frac) (iii) Re-fracturing (refracs) (iv) step-rate tests (v) shut-in (vi) periodic pumping test
.

After benchmarking the typical hydraulic features of the STIMTEC experiment (see section), Step 3 is to characterize the
in-situ hydraulic behavior, i.e. to identify the hydromechanical rock properties. The teams can choose their preferred fracture
network model, a suggested starting point is the main hydraulic feature as shown in Fig. 18.

### 2.3.4 Step 5: Synthesis and open sciences

Results will be synthesized for evaluation of numerical methods, model upscaling from lab to field scale, and applicability
of the methodology for related application areas (e.g. geothermal reservoirs in crystalline rock). SAFENET-2 will actively
contribute to open science action in nuclear waste management (Kolditz et al., 2023; Lehmann et al., 2024), e.g. by providing
benchmarking tools via an interactive web platform.

## 3 Conclusions

This paper provides a detailed description of SAFENET-2, a task of the new DECOVALEX-2027 project. After a short in-
troduction of the intention and concept of the task, the experimental facilities and basics are described. Rock samples will be
analyzed from underground research laboratories in Germany (Reiche Zeche) and China (Beishan). SAFENET-2 will again
follow two paths towards fully coupled thermo-hydro-mechanical (THM) fracture processes, (i) HM+T by extending HM with



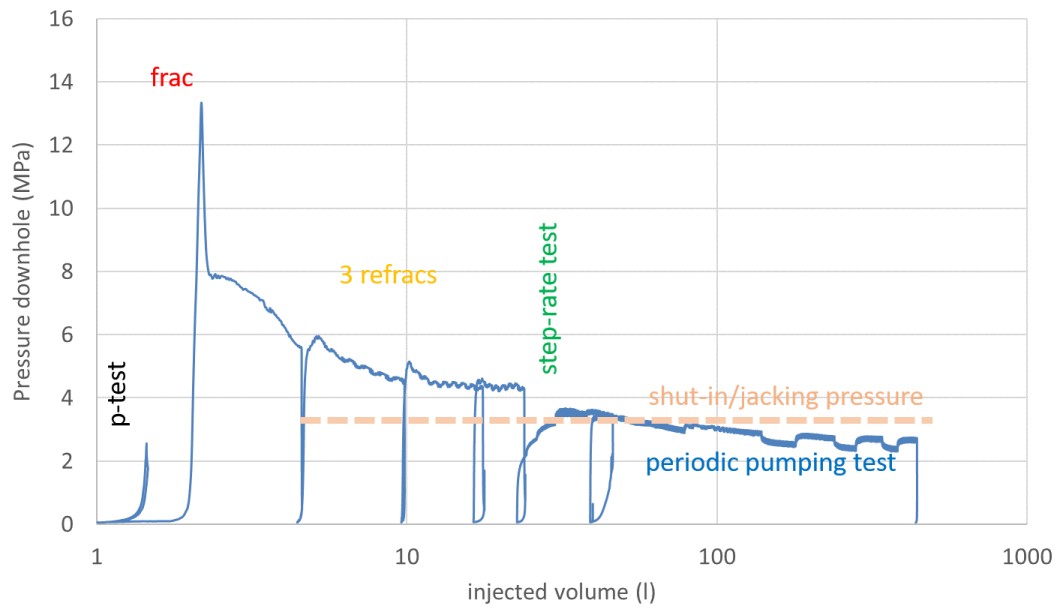

**Figure 17.** STIMTEC field experiment: Hydraulic in-situ testing (Boese et al., 2022, 2023)

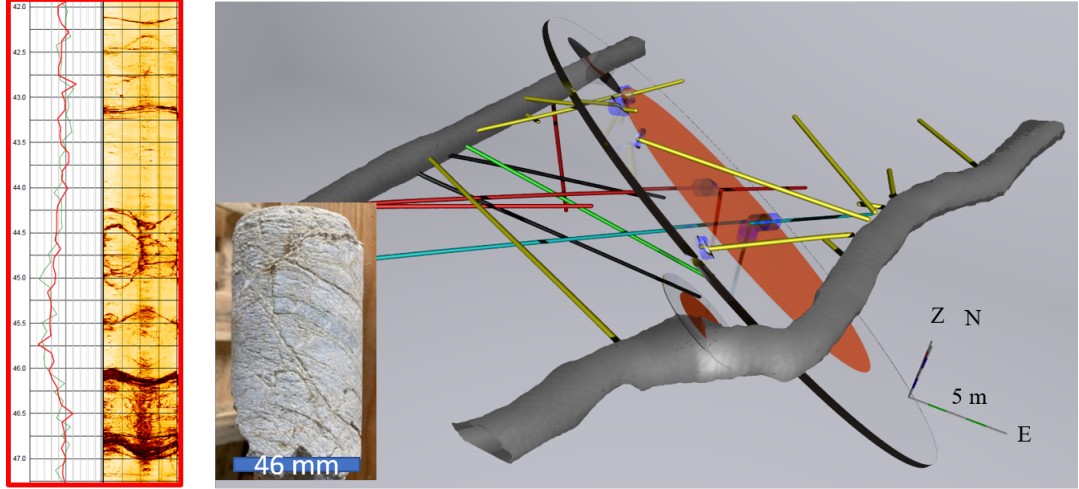

**Figure 18.** STIMTEC field experiment: Main hydraulic feature (Boese et al., 2022, 2023)

thermal processes and (ii) TM+H by extending thermo-mechanical with hydraulic processes. The numerical basis for fully coupled THM processes will be provided by a set of benchmarks for model and code testing. In addition, typical benchmarks will be designed to capture the main features of the new GREAT cell and thermoslip-flow experiments conducted at the Rock
Mechanics Laboratories in Edinburgh and Chongqing, respectively. Finally, the in-situ STIMTEC experiments are studied to



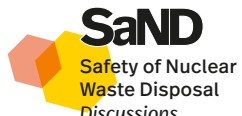

demonstrate the scalability of the models for field experiments. Fig. 2 shows the tentative time schedule for the SAFENET-2 Task.

This paper is dedicated to the special issue of SAND on "Trust-in-Models". Models will play an important role in the deep geological disposal of radioactive waste, as predictions of the possible evolution of repositories are a rigorous part of the safety assessment. Therefore, model validation is of utmost importance for the credibility of the modeling process in radioactive waste disposal. SAFENET contributes to this goal in several ways by (i) providing a new experimentally based benchmark suite for fracture models in crystalline rock, (ii) analyzing new unique experimental data for the description of THM processes in different crystalline rock samples, and (iii) applying validated models as a result of (i) and (ii) to the analysis of in-situ experiments at the field scale, thus also contributing to a better understanding of the upscaling behavior in crystalline rock masses. This paper presents the introduction and detailed description of the second SAFENET-2 task within the DECOVALEX 2027 project. The progress of the team's work will be reported in subsequent publications and in the synthesis paper with the main research results at the end of the task (Tab. 2).

| SAFENET-2 schedule | | | | 2024 | | 2025 | | 2026 | | 2027 | | |
|---|---|---|---|---|---|---|---|---|---|---|---|---|
| Step1 | | Benchmarking | | | | | | | | | | |
| | 1.1 | GREAT cell | | | | | | | | | | B1 |
| | 1.2 | Thermoslip-flow cell experiments | | | | | | | | | | B2 |
| | 1.3 | STIMTEC experiments | | | | | | | | | | B3 |
| | | Paper ideas | | | B1 | B2 | B3 | | | | | |
| Step2 | | GREAT experiments | | | | | | | | | | |
| | 2.1 | Hydraulic fracturing | | | | | | | | | | G1 |
| | 2.2a | Circulation flow experiment | | | | | | | | | | G2 |
| | 2.2b | Circulation flow experiment | | | | | | | | | | |
| | 2.3 | 3D version | | | | | | | | | | G3 |
| | | | | | | | G1 | G2 | G3 | | | |
| Step 3 | | Thermoslip-flow cell experiments | | | | | | | | | | |
| | 3.1 | M process | | | | | | | | | | |
| | 3.2 | TM process | | | | | | | | | | |
| | 3.3 | HM process | | | | | | | | | | |
| | 3.4 | THM process | | | | | | | | | | T1 |
| | 3.5 | 3D version | | | | | | | | | | T2 |
| | | Paper ideas | | | | | | | T1 | T2 | | |
| Step 4 | | STIMTEC experiments | | | | | | | | | | |
| | 4.1 | | | | | | | | | | | S1 |
| | | Paper ideas | | | | | | | | | S1 | |
| Step 5 | | Synthesis | | | | | | | | | | |
| | 5.1 | | | | | | | | | | | S2 |
| | | Paper ideas | | | | | | | | | S2 | |
| | | | | | | | | | | | | |
| | | Workshops | | WS1 | WS2 | WS3 | WS4 | WS5 | WS6 | WS7 | WS8 | |

**Table 2.** Time schedule for the SAFENET-2 Task



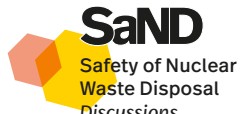

*Code availability.* SAFENET relies on both open source and commercial code. For the former, source codes are available from the respective websites. For both, the corresponding input files are made available on the DECOVALEX website.

*Data availability.* Experimental data are made available on the DECOVALEX website.

*Author contributions.* OK: research concept, benchmarking concept, main writing; CMCD: research concept, experimental concept and data, manuscript writing; JSY: research concept, numerical simulation; JR: experimental concept and data; LZ: experimental data, manuscript writing; AFH: experimental data; MC: experimental data; SG: experimental data; JW: rock samples and experimental data; MM: benchmarking concept, numerical simulation;

*Competing interests.* none

*Acknowledgements.* DECOVALEX (https://decovalex.org) is an international research project comprising participants from industry, government and academia, focusing on development of understanding, models and codes in complex coupled problems in subsurface geological and engineering applications; DECOVALEX-2027 is the current phase of the project. The authors appreciate and thank the DECOVALEX-2027 Funding Organisations Andra, BASE, BGE, BGR, CAS, CNSC, COVRA, US DOE, ENRESA, ENSI, JAEA, KAERI, NWMO, RWM,
SÚRAO, SSM and Taipower for their financial and technical support of the work described in this paper. The BGR is subordinate to the German Federal Ministry for Economic Affairs and Climate Action (BMWK). The statements made in the paper are, however, solely those of the authors and do not necessarily reflect those of the Funding Organizations. Furthermore, this work has been co-financed within the framework of EURAD, the European Joint Programme on Radioactive Waste Management (grant agreement No 847593). Financial support from the UK Engineering and Physical Sciences Research Council (EPSRC) for the project "Smart Pumping for Subsurface Engineering"
with grant number (EP/S005560/1) is gratefully acknowledged for the provision of experimental data relating to the GREAT cell.

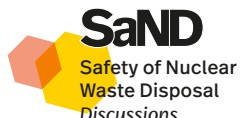

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
