# Peer review of "SAFENET - Fracture evolution in crystalline rocks (from lab to in-situ scale)"

_Safety of Nuclear Waste Disposal, 2024_

## Author Response (AR1)

**SAFENET-2: Manuscript revision**

First of all, we would like to thank the reviewers and the editor for the thorough review of the manuscript and the very helpful constructive comments, based on which the revised manuscript has been greatly improved.

In the following point-by-point responses to the reviewers' comments, the comments have been labelled as they appear in the manuscript, so that it is easy to locate the places where changes have been made in the manuscript revision. Text added to the manuscript is in blue.

**Reviewer #1**

This is a good basis for a project task like DECOVALEX-2027 SAFENET. It allows the general public to follow and even participate in this study. Therefore, this is a very good example of transparency for the public. However, as a scientific paper, it should be revised taking into account the following points:

| R1#1 | The introduction should describe work on the general progress of the granite study, in particular on the issue of deep disposal. It should answer the following questions:
Why is granite suitable for high-level waste disposal?
What are the main problems to be solved in this context?

Added: *Crystalline rocks are among the potential host rocks for nuclear waste repositories, particularly for the Nordic countries, Canada, Korea and Japan, which are rich in crystalline rock formations. A sound knowledge of the behaviour of crystalline rocks, in particular their strength as geological barriers, is of paramount importance. Therefore, the fracture mechanics of brittle rocks is the focus of SAFENET.* |
|---|---|
| R1#2 | What are the main outcomes of the past phases of DECOVALEX before D2023?

It will be very difficult to summarise the main results of the previous phases of DECOVALEX before D2023. We have added more references to review papers on DECOVALEX at the beginning of the Introduction. We have added a short summary of Safenet-1 (see R1#4) as suggested by the reviewer. |
| R1#3 | The experimental data are presented in a very heterogeneous manner. The experimental data in the section 2.2.1 are described in detail, but 2.2.2 and 2.2.3 are very limited without results.

We fully understand this comment. As the GREAT cell experiments for SAFENET-2 have been completed, this section already includes the most important experimental results. The thermoslip flow cell is now constructed and ready for experimental work, but the results will not be available until 2025, so it cannot be described in too much detail at this stage. STIMTEC will use the key stimulation experiment described in section 2.2.3 and referred to in the available publications. |
| R1#4 | Results from the previous phase D2023 should be summarized in one section.
The intention to compare such complex tests as benchmarking is good, but a big challenge. Would it be possible to summarized some numerical methods used in the previous phase?

Added: *The main results of SAFENET-1 are recently synthesized in \cite{Kolditz2024}. Safenet uses a systematic and experimental approach to numerically simulate mechanical (M), hydro-mechanical (HM) and thermo-mechanical (TM) fracture and now THM processes in brittle rocks. The task team has introduced, applied and compared a wide range of numerical methods, including both continuum and discontinuum methods, for simulating related fracture processes (e.g. FEM, DEM, cellular automata, numerical manifold method).* |

> *Experimental data of SAFENET-1 are based on three key experiments: the Freiberg, GREAT cell and KICT experiments, which analyse M, HM and TM processes respectively. Classic HM and THM benchmark exercises serve as a common basis, using analytical solutions for a plane-line discontinuity in a poroelastic medium \citep{Sneddon-Lowengrub-1969} and a point heat source in a thermo-poroelastic medium \citep{Booker1985173,Chaudhry20192743}. These solutions also serve as a reference for rough fractures and simple fracture networks.*
>
> *An analysis of the constant normal load (CNL) experiment was carried out using micro- and macroscopic approaches based on the Freiberg experiment. The GREAT cell experiments provided a database for evaluating the mechanical and hydromechanical responses of various rock samples (resin, greywacke, gneiss) in triaxial tests with a rotational stress field. Fracture permeability was determined as a function of normal stresses in the rotational stress field. The KICT experiments were used to investigate thermally induced shear slip and dilation processes.*

Review link: https://doi.org/10.5194/sand-2024-2-RC1

**Reviewer #2**

| | |
|---|---|
| | |
| R2#1 | The paper describes the planned activities in the context of SAFENET-2, a task of DECOVALEX-2027 project. The overall structure and level of detail of the paper is coherent with the initial stage of the activities. Yet, in some sections some additional clarifications are necessary to understand whether the described activities belong to SAFENET or SAFENET-2, and whether they have been concluded or not (detailed comments below).
 Once these aspects are clarified, I suggest the authors to reconsider the title of paper, and whether if should be explicitly refer to SAFENET-2 (or both, SAFENET part 1 and 2).

 The title has been changed to SAFENET-2 in order to clarify the purpose of the paper – an introduction to the new project phase. |
| R2#2 | 1.   In the abstract and conclusions, it is said that the paper discuss safenet-2 plan, yet there is no mention of safenet-2 in the introduction. The provided overview on Decovalex and Safenet is useful, but it is not clear where the first part of safenet finishes, and the second part (addressed here) begins. As an example, in line 50 "SAFENET will also elaborate the potential of Artificial Intelligence (AI) concepts for…" is this safenet 2 or some remaining activities in safenet?

 We use the name SAFENET to emphasise the common idea and basis of both phases of the project. We have checked the use of SAFENET-1 and SAFENET-2 throughout the paper to ensure that the completed first phase and the ongoing second phase are clearly identified. |
| | 2.   Section 2.1 Concept: |
| R2#3 | a.   References for the GREAT cell and the thermoslip-flow cell, line 60, are necessary

 References for both GREAT cell and thermoslip-flow cell have been added. |
| R2#4 | b.   The approaches mentioned in lines 60-61 needs clarifications. In the figure (and in the conclusions), two approaches are considered: HM+T and TM+H, meaning that temperature and hydraulic are added on top of pre-existing approaches. In the text a "HM" and "TM-H" processes are mentioned. Please make the notation consistent and consider rephrasing and adding few details on how thermal and hydraulic effects are going to be included.

 The notation has been made consistent and further short explanations have been added to make the reading easier. The experimental setup is described in detail in sections 2.2.1 and 2.2.2.

 The GREAT cell is additionally equipped with a heating device. The thermoslip-flow cell allows heating, triaxial loading and additional fluid injection into a fractured specimen. |
| R2#5 | c.   In line 64 "the second area of focus" is mentioned. Although before there was no mention of the number of areas of focus. Probably would be worth to either mention early on that there are two focuses, or rephrase here.

 SAFENET-2 will focus on two areas: firstly, the improvement of numerical models based on laboratory experiments. |
| | 3.   In section 2.2.1 GREAT cell large lab scale data for interpretation |
| R2#6 | a.   in this section, it is not clear whether all experiments planned for safenet 2 are already concluded, or the results mentioned here are just preliminary and other tests are planned. |

| | |
|---|---|
| | GREAT cell experiments are mainly available, new experiments are only planned on demand. |
| R2#7 | b.  In line 90, the "SAFENET-2 HM Task 2" is mentioned, yet it is not clear what is the task 2, as this doesn't seem to be mentioned before.

The Task structure is introduced in section 2.3 and additionally marked in Fig. 1 |
| R2#8 | c.  Figure 4 (and 6, 7): the unit of measurements represented is not clear. Are those µstrain? Doesn't it start from 0?

Yes, they are µstrain, or Strain x $10^{-6}$, and these can be positive and negative strain as they are relative measurements to a starting point. Starting at -100 allows the visual circular representation which should be more intuitive for the reader as a radar type plot of the strain around the surface of the sample. Most people will note that the points plot around the zero line. |
| R2#9: | d.  Figure 6 and 7: text in the figures is quite small and difficult to read.

Text in the figures is enlarged |
| R2#10 | e.  Line 138: "and" is repeated twice

corrected |
| R2#11 | f.  Line 148: it is not clear what the authors meant by "mean modelled" normal stress. Same for the "modelled" maximum shear stress in line 149.

The normal and shear stress on the fracture plane is not directly measured, but interpreted from the directional stress applied to the sample and the orientation of the fracture plane in this stress field. |
| R2#12 | g.  Line 148: after normal stress, a full stop is missing.

corrected |
| | 4.  Section 2.2.3 Stimtec in-situ experiments |
| R2#13 | a.  is this in situ experiments concluded? If so, it would be preferable to clarified that only the results would be exploited in the safenet-2, or else additional experimental activities should be detailed. In the planning at the end of paper, it seems that additional activities are planned for early next year.

STIMTEC experiments are finished and ready for interpretation. |
| R2#14 | b.  Line 175, the references should be without parenthesis

corrected |
| | 5.  Section 2.3 Modelling approach |
| R2#15 | a.  Line 179: Here the authors are probably referring to safenet-2. Please double check.

corrected |
| R2#16 | b.  Line 187: step 0 does not seem to have been described before.

corrected to Step 1 |
| R2#17 | c.  Line 191: should be methods and codes

corrected to codes |
| R2#18 | 6.  Section 2.3.3 Step 3: Thermoslip-flow test case simulation |
| | a.  The description at line 250 includes "keywords" of the test results, but it is not clear why the authors would mention keywords here. |

| | |
|---|---|
| | *Sentence has been rephrased.* |
| R2#19 | b.  Line 258-260: the authors refer to future workshops, which is unusual to me.  I'd leave the editor to judge whether this is acceptable for the current publication.

 *We would like to keep the workshop as this is were the material can be found.* |
| R2#20 | c.  Line 262: models "will be examined"

 *corrected* |
| R2#21 | d.  Line 267: full stop should be on the same line.

 *corrected* |
| | 7.  Section 3 Conclusions |
| R2#22 | a.  Line 286: the cross reference is to table 2 (instead of figure 3)

 *corrected to Tab. 2* |
| | |

Review link: https://doi.org/10.5194/sand-2024-2-RC2

---

## Author Response (AR2)

Final paper revision

Dear Carlo,
thank you for your final comments, which have been incorporated into the final version of the manuscript.
Best regards, Olaf